# CAN A BAYESIAN ORACLE PREVENT HARM FROM AN AGENT?

## ABSTRACT

Is there a way to design powerful AI systems based on machine learning methods that would satisfy probabilistic safety guarantees? With the long-term goal of obtaining a probabilistic guarantee that would apply in every context, we consider estimating a context-dependent bound on the probability of violating a given safety specification. Such a risk evaluation would need to be performed at runtime to provide a guardrail against dangerous actions of an AI. Noting that different plausible hypotheses about the world could produce very different outcomes, and because we do not know which one is right, we derive bounds on the safety violation probability predicted under the true but unknown hypothesis. Such bounds could be used to reject potentially dangerous actions. Our main results involve searching for cautious but plausible hypotheses, obtained by a maximization that involves Bayesian posteriors over hypotheses. We consider two forms of this result, in the i.i.d. case and in the non-i.i.d. case, and conclude with open problems towards turning such theoretical results into practical AI guardrails.

## 1 INTRODUCTION

Ensuring that an AI system will not misbehave is a challenging open problem (Bengio et al., 2024), particularly in the current context of rapid growth in AI capabilities. Governance measures and evaluation-based strategies have been proposed to mitigate the risk of harm from highly capable AI systems, but do not provide any form of safety guarantee when no undesired behavior is detected. In contrast, the *safe-by-design* paradigm involves AI systems with quantitative (possibly probabilistic) safety guarantees from the ground up, and therefore could represent a stronger form of protection (Dalrymple et al., 2024). However, how to design such systems remains an open problem too.

Since testing an AI system for violations of a safety specification in every possible context, *e.g.*, every (query, output) pair, is impossible, we consider a rejection sampling approach that declines a candidate output or action if it has a probability of violating a given safety specification that is too high. The question of defining the safety specification (the violation of which is simply referred to as "harm" below) is important and left to future work, possibly following up approaches such as constitutional AI (Bai et al., 2022). We also note that being Bayesian about the interpretation of a human-specified safety specification would protect against the AI wrongly believing an incorrect interpretation. Here we instead focus on a question inspired by risk-management practice (McNeil et al., 2015): even though the true probability of harm following from some proposed action is unknown, because the true data-generating process is unknown, can we bound that risk using quantities that can be estimated by machine learning methods given the observed data?

To illustrate this question, consider a committee of "wise" humans whose theories about the world are all equally compatible with the available data, knowing that an unknown member of the committee has the correct theory. Each committee member can make a prediction about the probability of future harm that would result from following some action in some context. Marginalizing this harm probability over the committee members amounts to making them vote with equal weights. If the majority is aligned with the correct member's prediction, then all is good, *i.e.*, if the correct theory predicts harm, then the committee will predict harm and can choose to avoid the harmful action. But what if the correct member is in the minority regarding their harm prediction? To get a *guarantee* that the true harm probability is below a given threshold, we could simply consider the committee member whose theory predicts the highest harm probability, and we would be sure that their harm probability prediction upper bounds the true harm probability. In practice, committee members are

not equally "wise", so we can correct this calculation based on how plausible the theory harbored by each committee member is. In a Bayesian framework, the plausibility of a theory corresponds to its posterior over all theories given the observed data, which is proportional to the data likelihood given the theory multiplied by the prior probability of that theory.

In this paper, we show how results about posterior consistency can provide probabilistic risk bounds. All the results have the form of inequalities, where the true probability of harm is upper bounded by a quantity that can in principle be estimated, given enough computational resources to approximate Bayesian posteriors over theories given the data. In addition, these are not hard bounds but only hold with some probability, and there is generally a trade-off between that probability and the tightness of the bound. We study two scenarios in the corresponding sections: the i.i.d. data setting in §3 and the non-i.i.d. data setting in §4, followed by an experiment in §5. In all cases, a key intermediate result is a bound relating the Bayesian posterior on the unknown true theory and the probability of other theories (with propositions labeled **True theory dominance**). The idea is that because the true theory generated the data, its posterior tends to increase as more data is acquired, and in the i.i.d. case it asymptotically dominates other theories. From such a relationship, the harm risk bound can be derived with very little algebra (yielding propositions labeled **Harm probability bound**).

We conclude this paper with a discussion of open problems that should be considered in order to turn such bounds into a safe-by-design AI system, taking into account the challenge of representing the notion of harm and reliable conditional probabilities, as well as the fact that, in general, the estimation of the required conditional probabilities will be imperfect.

**Related Work.** The concept of blocking actions based on probabilistic criteria resembles probabilistic shielding in Markov Decision Processes (MDPs) (Jansen et al., 2020), but our bounds do not require knowledge of the true model, extending beyond Carr et al. (2023)'s work on partially observable MDPs. While Beckers et al. (2023) and Richens et al. (2022) propose specific frameworks for quantifying harm, our approach remains agnostic, by only requiring harmful outcomes to be representable as binary events $H = 1$, allowing various harm definitions, while providing conservative probability bounds for safety-critical contexts. Osband & Van Roy (2017) study translating concentration bounds from a pure predictive setting to an MDP setting with exploration, whereas we address an orthogonal question: providing safety guarantees without relying on potentially harmful exploration to gain information.

## 2    SAFE-BY-DESIGN AI?

Before an AI is built and deployed, it is important that the developers have high assurances that the AI will behave well. Dalrymple et al. (2024) propose an approach to "guaranteed safe AI" designs with built-in high-assurance quantitative safety guarantees, although these guarantees can sometimes be probabilistic and only asymptotic. It remains an open question whether and how that research program can be realized. The authors take existing examples of quantitative guarantees in safety-critical systems and motivate why such a framework should be adopted if we ever build AI systems that match or exceed human cognitive abilities and could potentially act in dangerous ways. Their program is motivated by current known limitations of state-of-the-art AI systems based on deep learning, including the challenge of engineering AI systems that robustly act as intended (Cohen et al., 2022b; Krakovna et al., 2020; Pan et al., 2021; Pang et al., 2023; Zhuang & Hadfield-Menell, 2020; Skalse et al., 2022; 2023; Karwowski et al., 2023; Skalse et al., 2024).

The approach proposed by Dalrymple et al. (2024) has the following components: a *world model* (which can be a distribution about hypotheses explaining the data), a *safety specification* (what are considered unacceptable states of the world), and a *verifier* (a computable procedure that checks whether a policy or action violates the safety specification).

Here, we study a system that infers a probabilistic world model, or *theory*, $\tau$ and updates its estimate of $\tau$ via machine learning, using the stream of observed data $D$. The observations $D$ are assumed to come from a data-generating process given by a ground-truth world model $\tau^*$, which lies in the system's space of possible theories. The inference of the theory $\tau$ is Bayesian, meaning that the system maintains an estimate $q$ of the true posterior $P(- \mid D)$ over theories: $q(\tau \mid D) \approx P(\tau \mid D)$, where $P(\tau \mid D)$ is proportional to the product of the prior probability $P(\tau)$ with the likelihood of the observations under the theory, $P(D \mid \tau)$. In the simplest case, $q$ is a point estimate, optimally placing its mass on the mode of the posterior. Assuming an observation $x$ and a theory $\tau$ are independent

given $D$, inference of the latent theory $\tau$ allows the system to approximate conditional probabilities $P(y \mid x, D) \approx \mathbb{E}_{\tau \sim q(\tau \mid D)}[P_\tau(y \mid x, D)]$ over any random variables $X, Y$ known to the world model.

The safety specification is given in the form of a binary random variable $H$ (called "harm" below) whose probability given the other variables depends on the theory $\tau$. We are interested in predicting the probability of harm under the true theory $\tau^*$. Because $\tau^*$ is unknown, we propose to estimate upper bounds on this probability using the estimated posteriors. These upper bounds can be used as thresholds for a *verifier* that checks whether the risk of harm falls below some acceptable level.

Following Dalrymple et al. (2024), we assume that the notion of harm has been specified, possibly in natural language, and that the ambiguities about its interpretation are encoded within the Bayesian posterior $P(\tau \mid D)$. This paper focuses on the verifier under different assumptions of i.i.d. or non-i.i.d. data.

**What do the observations and context represent?**   We give a possible interpretation of the objects introduced in the preceding discussion in the simple case of an agent acting in a fully observed environment (Markov Decision Process, or MDP), where the theory is a transition model and the occurrence of harm at a state $s$ is conditionally independent of all other variables given $s$.

- Observations $Z$ are transitions $z = (s, a, s', r)$, where $s$ is a state, $a$ is an action, $s'$ is the next state, and $r$ is the reward received.
- Theories $\tau$ encode the state visitation, transition probabilities[1], as well as the behavior policy from which observations are collected.
- The dataset $D$ is a sequence of observed transitions.
  - In the non-i.i.d. setting, $D$ could consist, for example, of the observations from a finite rollout in the order in which they occurred.
  - In the i.i.d. setting, $D$ would need to be a sequence of independent samples from a fixed state-action-reward visit distribution. This could be achieved, for example, by rolling out a behavior policy multiple times and randomly sampling transitions from the resulting trajectories.
  In the common special case of a contextual bandit MDP under a fixed policy, the two coincide.
- The context $X$ is a pair $x = (s, a)$, where $s$ is a state and $a$ is an action being considered at state $s$.
- The harm probability $P(H = 1 \mid X = (s, a), \tau, D)$ can be any function of the theory $\tau$, the context $x = (s, a)$, and the data $D$. For example, this probability could be derived from a fixed specification of what it means for a state $s'$ to be harmful, $P_{\text{harm}}(H = 1 \mid s')$. Then, the harm probability could be computed as $P(H = 1 \mid X = (s, a), \tau, D) = \sum_{s', r} P_\tau(s', r \mid s, a) P_{\text{harm}}(H = 1 \mid s')$.

We note that the interpretation of harm probability in the example above includes the case where the occurrence of harm is an observed variable $s'_{\text{harm}}$ that is part of the state $s'$: in that case, we set $P_{\text{harm}}(H = 1 \mid s') = 1$ if $s'$ is harmful (*i.e.*, $s'_{\text{harm}} = 1$), and $P_{\text{harm}}(H = 1 \mid s') = 0$ otherwise. Then the harm probability is just the probability, under $\tau$, of reaching a harmful state, and observations of harm in $D$ affect the Bayesian posterior over theories.

This interpretation also includes the case where the harm probability is a function of the state $s'$, but (non)occurrence of harm is not observed in $D$. For example, a language model encoding world knowledge and human preferences or constraints, or an iterative reasoning procedure that uses those constraints, could generate some specification of harm $P_{\text{harm}}(H = 1 \mid s')$, perhaps unreliably.

Finally, a setting that separates the predicted next state $s'$ from the harm variable $H$ in this way gives a framework for studying how an agent might tamper with harm guardrails. If the state $s'$ decomposes as $s' = (s'_{\text{harm}}, s'_{\text{rest}})$, and $P_{\text{harm}}$ is deterministic as a function of $s'_{\text{harm}}$, except for some difficult-to-reach values of $s'_{\text{rest}}$, then the agent can try to reach those values of $s'_{\text{rest}}$, so that harm is 'recorded' as not having occurred, even though it has. We discuss this briefly at the end of §4.

## 3   I.I.D. DATA

Following the notation introduced in the previous section, here, we consider the easier-to-analyze case where the observed examples $D = (z_1, z_2, \ldots, z_n)$ are sampled i.i.d. from the unknown distribution $\tau^*$. Assuming that the prior assigns nonzero mass to $\tau^*$, and all theories are distinct distributions, it can be shown that the posterior $P(\tau \mid D)$ converges to a point mass at $\tau^*$. We show that for sufficiently large $n$, we can bound the probability under $\tau^*$ of a harm event $H = 1$ given conditions

---

[1]To be more precise, we can obtain transition probabilities from $\tau$ by conditioning on $(s, a)$ to get $P_\tau(s', r \mid s, a)$, but only for state-action pairs with non-zero probability under $\tau$.

$x$ (*e.g.*, a context and an action) by considering the probability of $H = 1$ given $x$ and $D$, under a plausible but "cautious" theory $\tilde{\tau}$ that maximizes $P(\tilde{\tau} \mid D) P(H = 1 \mid x, \tilde{\tau}, D)$.

**Setting.** Fix a complete separable metric space $\mathcal{Z}$, called the *observation space*, let $\mathcal{F}$ be its Borel $\sigma$-algebra, and fix a $\sigma$-finite measure $\mu$ on $\mathcal{F}$. A *theory* is a probability distribution on the measurable space $(\mathcal{Z}, \mathcal{F})$ that is absolutely continuous w.r.t. $\mu$. If $\tau$ is a theory, we denote by $P_\tau(\cdot)$ the Radon-Nikodym derivative $\frac{d\tau}{d\mu} : \mathcal{Z} \to \mathbb{R}_{\geq 0}$, which is uniquely defined up to $\mu$-a.e. equality.

One can keep in mind two cases:

(1) $\mathcal{Z}$ is a finite or countable set and $\mu$ is the counting measure. Theories $\tau$ are equivalent to probability mass functions $P_\tau : \mathcal{Z} \to \mathbb{R}_{\geq 0}$.
(2) $\mathcal{Z} = \mathbb{R}^d$ and $\mu$ is the Lebesgue measure. Theories are equivalent to their probability density functions $P_\tau : \mathcal{Z} \to \mathbb{R}_{\geq 0}$ up to a.e. equality.

Consider a countable (possibly finite) set of theories $\mathcal{M}$ containing a *ground truth* theory $\tau^*$ and fix a choice of a (measurable) density function $P_\tau$ for each $\tau \in \mathcal{M}$.

**Definition of posterior as a random variable.** If $P$ is a prior distribution[2] on $\mathcal{M}$ and $z \in \mathcal{Z}$, we define the posterior to be the distribution with mass function

$$P(\tau \mid z) = \frac{P(\tau) P_\tau(z)}{\sum_{\tau' \in \mathcal{M}} P(\tau') P_{\tau'}(z)} \propto P(\tau) P_\tau(z), \tag{1}$$

assuming the denominator converges and the sum is nonzero. Otherwise, the posterior is considered to be undefined. As written, the posterior depends on the choice of density functions $P_\tau$, but any two $P_\tau$ that are $\mu$-a.e. equal yield the same posterior for $\mu$-a.e. $z$.

For $z_1, z_2 \in \mathcal{Z}$, we write $P(\cdot \mid z_1, z_2)$ for the posterior given observation $z_2$ and prior $P(\cdot \mid z_1)$, and similarly for a longer sequence of observations. It can be checked that $P(\cdot \mid z_1, \ldots, z_t)$ is invariant to the order of $z_1, \ldots, z_t$ and that it is defined in one order if and only if it is defined in all orders. This allows us to unambiguously write $P(\cdot \mid D)$ where $D$ is a finite multiset of observations, and we have

$$P(\tau \mid D) \propto P(\tau) \prod_{z \in D} P_\tau(z). \tag{2}$$

Let $\tau^* \in \mathcal{M}$ be the ground truth theory and $P(\cdot)$ a prior over $\mathcal{M}$. Consider a sequence of i.i.d. $\mathcal{Z}$-valued random variables $Z_1, Z_2, \ldots$ (whose realizations are the *observations*), where each $Z_i$ follows the distribution $\tau^*$. For any $t \in \mathbb{N}$, the posterior $P(\cdot \mid Z_{1:t})$ is then a random variable taking values in the space of probability mass functions on $\mathcal{M}$.[3]

**Bayesian posterior consistency.** We recall and state, in our setting, a result about the concentration of the posterior at the ground truth theory $\tau^*$ as the number of observations increases.

**Proposition 3.1** (**True theory dominance**)**.** *Under the above conditions and supposing that* $P(\tau^*) > 0$*, the posterior* $P(\cdot \mid Z_{1:t})$ *is almost surely defined for all n, and the following almost surely hold:*

*(a)* $P(\cdot \mid Z_{1:t}) \xrightarrow{t \to \infty} \delta_{\tau^*}$ *as measures, where* $\delta_{\tau^*}$ *is the Dirac measure, which assigns mass* 1 *to the theory* $\tau^*$ *and* 0 *elsewhere; equivalently,* $\lim_{t \to \infty} P(\tau \mid Z_{1:t}) = \mathbb{1}[\tau = \tau^*]$.
*(b) There exists* $N \in \mathbb{N}$ *such that* $\arg\max_{\tau \in \mathcal{M}} P(\tau \mid Z_{1:t}) = \tau^*$ *for all* $t \geq N$.

(All proofs can be found in appendix A.) Note that this result assumes that all theories in $\mathcal{M}$ are distinct *as probability measures* (so no two of the $P_\tau$ are $\mu$-a.e. equal).

**On necessity of conditions.** The i.i.d. assumption in Prop. 3.1 is necessary; see Remark 4.3 for an example where $\limsup_{t \to \infty} P(\tau^* \mid Z_{1:t})$ does not almost surely approach 1.

**Remark 3.2.** *The assumption that the data-generating process* $\tau^*$ *lies in* $\mathcal{M}$ *and has positive prior mass is also necessary for convergence of the posterior. To illustrate this, we give a simple ex-*

---

[2]To be precise, $\mathcal{M}$ is endowed with the counting measure and we flexibly interchange distributions and mass functions on $\mathcal{M}$.

[3]To be precise, if the $Z_i$'s are measurable functions from a sample space $\Omega$ to $\mathcal{Z}$ and $\langle Z_1, \ldots, Z_t \rangle$ is their pairing, the random variable $P(\cdot \mid Z_{1:t}) : \Omega \xrightarrow{\langle Z_1, \ldots, Z_t \rangle} \mathcal{Z}^t \xrightarrow{P(\cdot \mid -)} \mathbb{P}(\mathcal{M})$ has codomain the space $\mathbb{P}(\mathcal{M})$ of functions $\mathcal{M} \to \mathbb{R}_{\geq 0}$ summing to 1. The function $P(\cdot \mid -)$ mapping a sequence of observations to the posterior probability mass function is measurable, due to each $P_\tau(z)$ being measurable in $z$ and elementary facts.

*ample in which the theories are Bernoulli distributions and the posterior does not converge to any distribution over $\mathcal{M}$.*

*Take $\mathcal{Z} = \{-1, 1\}$ and $\mathcal{M} = \{\tau_p, \tau_{1/2}, \tau_{1-p}\}$ for some $\frac{1}{2} < p < 1$, where $P_{\tau_c}(1) = c$. Assume a prior with $P(\tau_p) = P(\tau_{1-p}) = \frac{1}{2}$ and take the true data-generating process $\tau^*$ to be $\tau_{1/2}$, which has prior mass 0. The log-ratio of posterior masses is then an unbiased random walk:*

$$\log \frac{P(\tau_p \mid Z_{1:t})}{P(\tau_{1-p} \mid Z_{1:t})} = \log \frac{P_{\tau_p}(Z_{1:t})}{P_{\tau_{1-p}}(Z_{1:t})} = \left( \log \frac{p}{1-p} \right) \sum_{i=1}^t Z_i.$$

*This quantity almost surely takes on arbitrarily large and small values infinitely many times. In fact, by the law of iterated logarithms, for any $\epsilon > 0$ there are infinitely many $t$ such that*

$$\log \frac{P(\tau_p \mid Z_{1:t})}{P(\tau_{1-p} \mid Z_{1:t})} \geq (1 - \epsilon) \left( \log \frac{p}{1-p} \right) \sqrt{2t \log \log t}$$

*and the same holds for $\log \frac{P(\tau_{1-p} \mid Z_{1:t})}{P(\tau_p \mid Z_{1:t})}$, by symmetry. In particular, $\lim_{t \to \infty} P(\tau \mid Z_{1:t})$ almost surely does not exist for any $\tau \neq \tau^*$, and the $\liminf$ and $\limsup$ are almost surely 0 and 1, respectively.*

However, in practice, Prop. 3.1 can be adapted to more general scenarios, by substituting the subset $\mathcal{T} \subseteq \mathcal{M}$ of theories with minimum relative entropy to $\tau^*$ for $\tau^*$ (when $\tau^*$ is not in $\mathcal{M}$). Then, we can replace convergence to $\delta_{\tau^*}$ with $P(\mathcal{T} \mid Z_{1:t}) \to 1$ in (a), and replace $\tau^*$ with $\mathcal{T}$ in (b).

**On generalizations to uncountable sets of theories.** We have critically used that the set of theories $\mathcal{M}$ is countable in the proof above when passing from almost sure convergence under $\tau^*$ sampled from the prior to almost sure convergence for any particular $\tau^*$ with positive prior mass. This argument fails for uncountable $\mathcal{M}$; indeed, characterization of the $\tau^*$ for which the posterior converges to $\delta_{\tau^*}$ is a delicate problem (see, *e.g.*, (Freedman, 1963; 1965; Diaconis & Freedman, 1986)). Concentration of the posterior in neighborhoods of $\tau^*$ under some topology on $\mathcal{M}$ has been studied by Schwartz (1965); Barron et al. (1999); Miller (2021), among others. For *parametric* families of theories with parameter $\theta \in \mathbb{R}^d$, under smoothness and nondegeneracy assumptions, the Bernstein-von Mises theorem guarantees convergence of the posterior $P(\theta \mid Z_{1:t})$ to the true parameter $\theta^*$ at a rate that is asymptotically Gaussian with inverse covariance $I(\theta^*)t$, where $I(\cdot)$ denotes the Fisher information matrix.

**On convergence rates.** While we do not handle the *rate* of convergence in Prop. 3.1, guarantees can be obtained under specific assumptions on the prior and the set of theories.

For example, for any $\tau \in \mathcal{M}$, the quantity $D_\tau^t := \log \frac{P(\tau^* \mid Z_{1:t})}{P(\tau \mid Z_{1:t})}$ is a process with $D_\tau^0 = \log \frac{P(\tau^*)}{P(\tau)}$ and i.i.d. increments, with

$$\mathbb{E}[D_\tau^{t+1} - D_\tau^t] = D_{KL}(\tau^* \parallel \tau), \quad \mathbb{E}\left[ (D_\tau^{t+1} - D_\tau^t)^2 \right] = \mathbb{E}_{Z \sim \tau^*} \left[ \left( \log \frac{P_{\tau^*}(Z)}{P_\tau(Z)} \right)^2 \right]. \tag{3}$$

Under the assumption that the variances are finite and uniformly bounded in $\tau$, the central limit theorem would give posterior convergence rate guarantees.

Note that, above, we make no assumptions on the theories, and Prop. 3.1 is a 'law-of-large-numbers-like' result that holds even if the variances in (3) are not finite and uniformly bounded.

**Harm probability bounds.** So far, we have considered a collection $\mathcal{M}$ of distributions over an observation space. Now, we show bounds when each theory computes probabilities over some additional variables. The following lemma extends Prop. 3.1 (b) to estimates of real-valued functions of the theories and observations.

**Lemma 3.3.** *Under the conditions of Prop. 3.1, let $f : \mathcal{M} \times \bigcup_{t=0}^\infty \mathcal{Z}^t \to \mathbb{R}_{\geq 0}$ be a bounded measurable function. Then there exists $N \in \mathbb{N}$ such that for all $t \geq N$ and any $\tilde{\tau} \in \arg\max_\tau [P(\tau \mid Z_{1:t}) f(\tau, Z_{1:t})]$, it holds that $f(\tau^*, Z_{1:t}) \leq f(\tilde{\tau}, Z_{1:t})$.*

A particular case of interest is when each theory is associated with estimates of probabilities of harm ($H = 1$) given a context $x$ and past observations $Z_{1:t}$. That is, $\mathcal{M}$ gives rise to a collection of conditional probability mass functions over the possible harm outcomes, denoted $P(\cdot \mid x, \tau, Z_{1:t})$, for every $x$ lying in some space of possible contexts. In this setting, we have the following corollary:

**Proposition 3.4** (**Harm probability bound**). *Under the same conditions as Prop. 3.1, there exists $N \in \mathbb{N}$ such that for all $t \geq N$ and $\tilde{\tau} \in \arg\max_\tau \mathrm{P}(\tau \mid Z_{1:t}) \mathrm{P}(H = 1 \mid x, \tau, Z_{1:t})$, it holds that*

$$\mathrm{P}(H = 1 \mid x, \tau^*, Z_{1:t}) \leq \mathrm{P}(H = 1 \mid x, \tilde{\tau}, Z_{1:t}). \tag{4}$$

## 4  NON-I.I.D. DATA

In this section, we remove the assumption made in §3 that the $Z_i$'s are i.i.d. given a theory $\tau^*$.

**Setting.**  As before, let $(\mathcal{Z}, \mathcal{F}, \mu)$ be a $\sigma$-finite Borel measure space of observations. For the results below to hold, we must also assume that $(\mathcal{Z}, \mathcal{F})$ is a Radon space (*e.g.*, any countable set or manifold), so as to satisfy the conditions of the disintegration theorem.

Let $(\mathcal{Z}^\infty, \mathcal{F}^\infty, \mu^\infty)$ be the space of infinite sequences of observations, $\mathcal{Z}^\infty = \{(z_1, z_2, \dots) : z_i \in \mathcal{Z})\}$, with the associated product $\sigma$-algebra and $\sigma$-finite measure. This object is the projective limit of the measure spaces $(\mathcal{Z}^t, \mathcal{F}^{\otimes t}, \mu^{\otimes t})$, where $\mathcal{Z}^t = \{(z_1, \dots, z_t) : z_i \in \mathcal{Z}\}$ and the projection $\mathcal{Z}^{t+1} \to \mathcal{Z}^t$ 'forgets' the observation $z_{t+1}$. A *theory* $\tau$ is a probability distribution on $(\mathcal{Z}^\infty, \mathcal{F}^\infty)$ that is absolutely continuous w.r.t. $\mu^\infty$. For $A \in \mathcal{F}^{\otimes t}$, we write $\tau_{1:t}(A)$ for the measure of the cylindrical set, $\tau(A \times \mathcal{Z} \times \mathcal{Z} \times \dots)$, so $\tau_{1:t}$ is a measure on $(\mathcal{Z}^t, \mathcal{F}^{\otimes t})$. Because $\mathcal{F}^\infty$ is generated by cylindrical sets, the absolute continuity condition on $\tau$ is equivalent to absolute continuity of $\tau_{1:t}$ w.r.t. $\mu^{\otimes t}$ for all $t$.[4] This condition allows to define measurable probability density functions $\mathrm{P}_\tau : \mathcal{Z}^t \to \mathbb{R}_{\geq 0}$ as Radon-Nikodym derivatives, so that

$$\forall A \in \mathcal{F}^{\otimes t}, \quad \tau_{1:t}(A) = \int_{z_{1:t} \in A} \mathrm{P}_\tau(z_{1:t}) \, d\mu^{\otimes t},$$

and measurable conditional probability densities $\mathrm{P}_\tau(z_{t+1} \mid z_{1:t}) := \frac{\mathrm{P}_\tau(z_{1:t}, z_{t+1})}{\mathrm{P}_\tau(z_{1:t})}$ when $\mathrm{P}_\tau(z_{1:t}) > 0$. The disintegration theorem for product measures implies that these conditionals and marginals over finitely many observations can be manipulated algebraically using the usual rules of probability for $\mu^\infty$-a.e. collection of values, *e.g.*, one has the autoregressive decomposition $\mathrm{P}_\tau(z_{1:t}) = \prod_{i=1}^t \mathrm{P}_\tau(z_i \mid z_{1:i-1})$, with the conditional $\mathrm{P}_\tau(z_1 \mid z_{1:0})$ understood to be the marginal $\mathrm{P}_\tau(z_1)$.

A theory is canonically associated with a random variable $Z_{1:\infty}$ taking values in $\mathcal{Z}^\infty$. We denote its components by $Z_1, Z_2, \dots$ and the collection of the first $t$ observations by $Z_{1:t}$.

**Definition of posterior as a random variable.**  Let $\mathcal{M} = (\tau_i)_{i \in I}$ be a collection of theories indexed by a countable set $I$[5] and let $\mathrm{P}$ be a prior distribution on $I$. We define the posterior over indices to be

$$\mathrm{P}(i \mid z_{1:t}) := \frac{\mathrm{P}(i) \, \mathrm{P}_{\tau_i}(z_{1:t})}{\sum_{j \in I} \mathrm{P}(j) \, \mathrm{P}_{\tau_j}(z_{1:t})}, \tag{5}$$

assuming the denominator converges to a positive value.

Consider a ground truth index $i^* \in I$ and abbreviate $\tau^* := \tau_{i^*}$. Let $Z_{1:\infty}$ be the random variable taking values in $\mathcal{Z}^\infty$ corresponding to $\tau^*$. Similarly to the i.i.d. case, the posterior $\mathrm{P}(\cdot \mid Z_{1:t})$ is a random variable taking values in the space of probability mass functions on $I$.

For all results below, we assume that $\mathrm{P}(i^*) > 0$.

**Bayesian posterior convergence.**  Previous work (*e.g.*, (Cohen et al., 2022a)) has shown that if $Z_{1:\infty} \sim \tau^*$, then the limit inferior of $\mathrm{P}(i^* \mid Z_{1:t})$ is almost surely positive. More generally, with probability at least $1 - \delta$, the posterior on the truth will not asymptotically go below $\delta$ times the prior on the truth. We repeat that result here in our notation.

**Lemma 4.1** (Martingale). *The process $W_t := \mathrm{P}(i^* \mid Z_{1:t})^{-1}$ is a supermartingale, i.e., it does not increase over time in expectation.*

**Proposition 4.2** (Posterior on truth). *For all $\delta > 0$, with probability at least $1 - \delta$, $\inf_t \mathrm{P}(i^* \mid Z_{1:t}) \geq \delta \mathrm{P}(i^*)$; that is, $\tau^*\big(\{z_{1:\infty} : \inf_t \mathrm{P}(i^* \mid z_{1:t}) < \delta \mathrm{P}(i^*)\}\big) \leq \delta$, or equivalently:*

$$\tau^*\big(\sup_{t \geq 0} W_t \geq (\delta \mathrm{P}(i^*))^{-1}\big) \leq \delta \tag{6}$$

---

[4]This is, in turn, equivalent to absolute continuity of conditional distributions, *i.e.*, for every measurable subset $A \subseteq \mathcal{Z}^t$ such that $\tau_{1:t}(A) > 0$, $\frac{1}{\tau_{1:t}(A)} \tau_{1:t+1}|_{A \times \mathcal{Z}} \ll \mu^{t+1}|_{A \times \mathcal{Z}}$, where $A \times \mathcal{Z} \subseteq \mathcal{Z}^t \times \mathcal{Z} \cong \mathcal{Z}^{t+1}$.

[5]Unlike in §3, we do not require theories to be distinct for the results in this section.

In the language of financial markets, if $W_t$ was the price of a stock at time $t$, you could never make money in expectation by holding it. Suppose that you "bought shares" at time 0, paying $W_0$, and waited for their value to increase by a factor of $\delta^{-1}$. If (6) did not hold and the probability of such an increase occurring was greater than $\delta$, then you could make an expected profit by "$\delta^{-1}$-tupling" your money with probability greater than $\delta$.

**Remark 4.3.** *Prop. 4.2 is "tight" in the following sense: for all $\delta, \varepsilon > 0$, there exist $\mathcal{M}$, P, and $\tau_{i^*} \in \mathcal{M}$, such that with probability at least $\delta$, $\limsup_t \mathrm{P}(i^* \mid Z_{1:t}) \leq (\delta + \varepsilon)\,\mathrm{P}(i^*)$.*

*We construct such an example. Consider the following setting: $\mathcal{M} = \{\tau^*, \tau'\}$ (indexed by $I = \{i^*, i'\}$ as $\tau_{i^*} = \tau^*, \tau_{i'} = \tau'$), $\mathcal{Z} = \{0, 1\}$, and the theories are defined by*

$$\mathrm{P}_{\tau^*}(1) = \delta, \quad \mathrm{P}_{\tau'}(1) = 1, \quad \mathrm{P}_{\tau_i}(1 \mid z_{1:t}) = \frac{1}{2} \; \forall i \in I, t \geq 1, z_{1:t} \in \mathcal{Z}^t.$$

*One has $\mathrm{P}(i^* \mid Z_1 = 1) = \frac{\delta\,\mathrm{P}(i^*)}{\delta\,\mathrm{P}(i^*) + \mathrm{P}(i')} < \delta\frac{\mathrm{P}(i^*)}{1 - \mathrm{P}(i^*)}$. Since $\tau^*$ and $\tau'$ give the same conditional probabilities of $Z_t$ given $Z_{1:t-1}$ for $t > 1$, one has $\mathrm{P}(i^* \mid Z_{1:t}) = \mathrm{P}(i^* \mid Z_1)$. So, for all $t \geq 1$, $\mathrm{P}(i^* \mid Z_1 = 1, Z_{2:t} = z_{2:t}) < \delta(1 - \mathrm{P}(i^*))^{-1}\,\mathrm{P}(i^*)$, and hence*

$$\tau^*\left(\left\{z_{1:\infty} : \limsup_t \mathrm{P}(i^* \mid z_{1:t}) < \delta(1 - \mathrm{P}(i^*))^{-1}\,\mathrm{P}(i^*)\right\}\right) \geq \tau^*\left(\left\{z_{1:\infty} : z_1 = 1\right\}\right) = \mathrm{P}_{\tau^*}(1) = \delta.$$

*So by choosing $\mathrm{P}(i^*) < 1 - 1/(1 + \frac{\varepsilon}{\delta})$, so that $\delta(1 - \mathrm{P}(i^*))^{-1} < \delta + \varepsilon$, we get the desired property.*

**Harm probability bounds.** We now state analogues of Prop. 3.4 in the non-i.i.d. setting. As above, let $H_t$ be a binary random variable that may depend on $Z_{1:t}, \tau$, and a context variable $x_t$.

**Proposition 4.4** (Weak harm probability bound). *For any $\delta > 0$, with probability at least $1 - \delta$, the following holds for all $t \in \mathbb{N}$ and all $x_t$:*

$$\mathrm{P}(H_t = 1 \mid Z_{1:t}, \tau^*, x_t) \leq \sup_{i \in I} \frac{\mathrm{P}(i \mid Z_{1:t})\,\mathrm{P}(H_t = 1 \mid Z_{1:t}, \tau_i, x_t)}{\delta\,\mathrm{P}(i^*)}.$$

Next, we show how the bound in Prop. 4.4 can be strengthened by restricting to theories that have sufficiently high posterior mass relative to theories that are "better" than them.

Let $i_{Z_{1:t}}^1, i_{Z_{1:t}}^2, i_{Z_{1:t}}^3, \ldots$ be an enumeration of $I$ in order of decreasing posterior weight $\mathrm{P}(i \mid Z_{1:t})$, breaking ties arbitrarily, for example, following some fixed enumeration of $I$ (*i.e.*, we have $\mathrm{P}(i_{Z_{1:t}}^n \mid Z_{1:t}) \geq \mathrm{P}(i_{Z_{1:t}}^{n+1} \mid Z_{1:t})$ for all $n$). Each $i_{Z_{1:t}}^n$ is an $I$-valued random variable (*i.e.*, the index of a theory in $\mathcal{M}$). For any $0 < \alpha \leq 1$, we also define the $\mathcal{P}(I)$-valued random variable

$$\mathcal{I}_{Z_{1:t}}^\alpha := \left\{ i_{Z_{1:t}}^n \in I : \mathrm{P}(i_{Z_{1:t}}^n \mid Z_{1:t}) \geq \alpha \sum_{m \leq n} \mathrm{P}(i_{Z_{1:t}}^m \mid Z_{1:t}) \right\}, \tag{7}$$

which is the set of indices that contain at least $\alpha$ of the posterior mass of all indices that are more likely than it under the posterior. If $\alpha = 1$, this set is the singleton $\{i_{Z_{1:t}}^1\}$. For any $0 < \alpha < 1$, it is nonempty, because it contains $i_{Z_{1:t}}^1$, and finite, since $|\mathcal{I}_{Z_{1:t}}^\alpha| \geq N$ implies (easily by induction) that $\sum_{i \in \mathcal{I}_{Z_{1:t}}^\alpha} \mathrm{P}(i \mid Z_{1:t}) \geq \left(\frac{1}{1-\alpha}\right)^{N-1} \mathrm{P}(i_{Z_{1:t}}^1 \mid Z_{1:t})$.

The following proposition is a variant of Cohen et al. (2022a, Thm 2).

**Proposition 4.5** (**True theory dominance**). *If $\alpha < \delta\,\mathrm{P}(i^*)$, then with probability at least $1 - \delta$, for all $t \in \mathbb{N}$, $i^* \in \mathcal{I}_{Z_{1:t}}^\alpha$.*

**Proposition 4.6** (**Harm probability bound**). *If $\alpha < \delta\,\mathrm{P}(i^*)$, then with probability at least $1 - \delta$, for all $t \in \mathbb{N}$ and all $x_t$,*

$$\mathrm{P}(H_t = 1 \mid Z_{1:t}, \tau^*, x_t) \leq \max_{i \in \mathcal{I}_{Z_{1:t}}^\alpha} \mathrm{P}(H_t = 1 \mid Z_{1:t}, \tau_i, x_t) \tag{8}$$

Because the conclusion of Prop. 4.6 is much stronger than that of Prop. 4.4, it would be much safer (or more useful, depending on the value of $\alpha$) to use $\arg\max_{i \in \mathcal{I}_{Z_{1:t}}^\alpha} \mathrm{P}(H_t = 1 \mid Z_{1:t-1}, \tau_i, x_t)$ as a 'paranoid' theory rather than $\arg\max_{i \in I} \mathrm{P}(\tau_i \mid Z_{1:t-1})\,\mathrm{P}(H_t = 1 \mid Z_{1:t-1}, \tau_i, x_t)$. The factor of $(\delta\,\mathrm{P}(i^*))^{-1}$ in Prop. 4.4 could render the upper bound on harm probability much larger than the trivial upper bound of 1. However, we note that approximating $\mathcal{I}_{Z_{1:t}}^\alpha$ – such as by amortization or by Monte Carlo methods – is much more difficult than approximating the posterior alone.

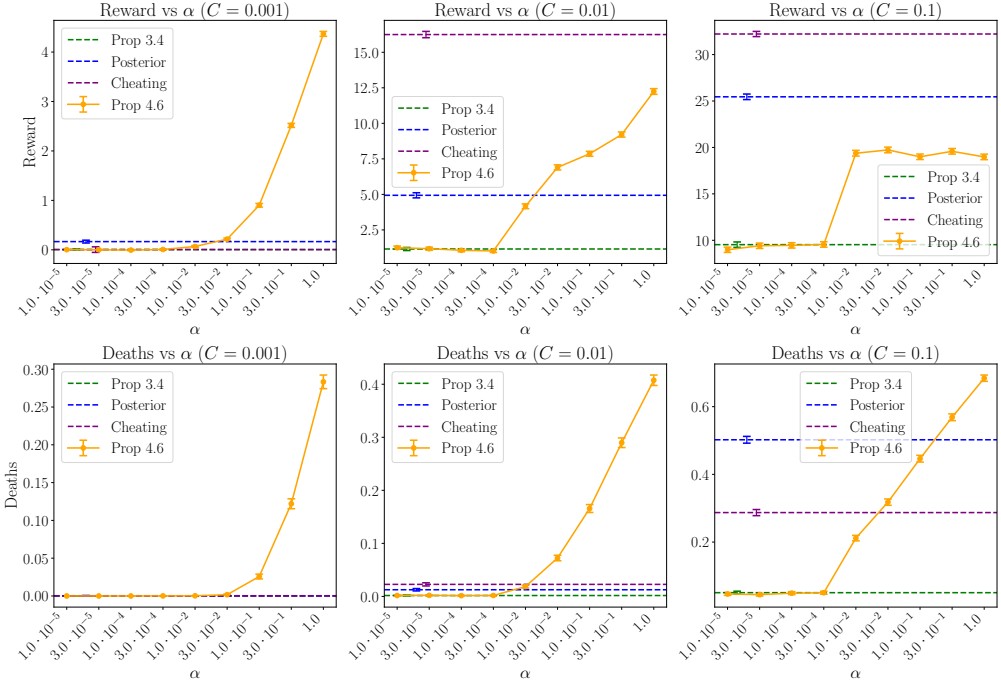

Figure 1: Mean episode deaths and reward for different guardrails in the exploding bandit setting.

**On the harm-recording mechanism.** Suppose that $\tau_{i^*} = \tau^*$ is a data-generating process meeting the description "$H_t = 1$ when harm has occurred", while $\tau_{i^\dagger} = \tau^\dagger$ is a data-generating process meeting the description "$H_t = 1$ when harm is recorded as having occurred" and agreeing with $\tau^*$ in its observational predictions otherwise. If, and only if, the recording process is functioning correctly, $\tau^* = \tau^\dagger$. For as long as the recording process is functioning correctly, $P(i^* \mid Z_{1:t})/P(i^\dagger \mid Z_{1:t}) = P(i^*)/P(i^\dagger)$. If the recording process ever fails at time $t$, then $Z_t \sim P_{\tau^\dagger}$, not $P_{\tau^*}$, since $Z_t$ is the result of this recording process; therefore, $P(i^* \mid Z_{1:t})/P(i^\dagger \mid Z_{1:t})$ would *decrease* in expectation, perhaps dramatically. We should not expect $P(i^*)$ to naturally win out over $P(i^\dagger)$, even if there are no mistakes when recording how harmful certain situations are. However, the following holds with probability approaching 1 as $\alpha \to 0$: for all $t$, if the recording process has not failed by time $t$, $\mathcal{I}_{Z_{1:t}}^\alpha$ contains both $i^*$ and $i^\dagger$. If $\tau^*$ considers tampering with the recording process to be a 'harmful' outcome, then an AI system could attempt to avoid a first instance of tampering at time $t$, for all $t$.

## 5 EXPERIMENTS

**Exploding bandit setting.** We evaluate the performance of safety guardrails based on Prop. 3.4 and Prop. 4.6 in a bandit MDP with 10 arms (actions). Each arm $a \in \{1, \ldots, 10\}$ is represented by a feature vector $f_a \in \{0, 1\}^d$ (we take $d = 10$, but $d$ is not necessarily equal to the number of arms), which is sampled uniformly at random at the start of each episode and known to the agent. The reward distribution of each arm is fixed for the duration of each episode and assumed to be of the following form: the reward received after taking action $a$ follows a unit-variance normal distribution, $r(a) \sim \mathcal{N}(f_a \cdot v^*, 1)$, where $v^* \in \{0, 1\}^d$ is some vector sampled uniformly at random at the start of each episode and unknown to the agent. Taking any action and observing the reward gives evidence about the identity of $v^*$ and thus about the reward distributions of the other actions. The agent maintains a belief over the vector used to compute the reward, beginning with a uniform prior over $\{0, 1\}^d$ and updating its posterior with each observation of an action-reward pair.

We assume that the agent samples its actions from a Boltzmann policy (with temperature 2) using the expected reward of each action under its posterior given the data seen so far, meaning that a reward vector $v \in \{0, 1\}^d$ determines a distribution over sequences of action-reward pairs. Thus each $v \in \{0, 1\}^d$ can be naturally associated with a theory $\tau_v$[6], and thus $I := \{0, 1\}^d$ is an indexing set for a collection of theories $\mathcal{M} = (\tau_v)_{v \in I}$. Inference of $v$ with evidence collected on-policy is equivalent to inference of $\tau_v$ given data generated by a true theory $\tau^* := \tau_{v^*}$. Since the policy changes across timesteps, so does the distribution of action-reward pairs, so we are in the non-i.i.d. setting.

---

[6]The mapping $v \mapsto \tau_v$ is not necessarily injective – multiple vectors may represent the same collection of reward distributions and therefore the same distribution over sequences of action-reward pairs.

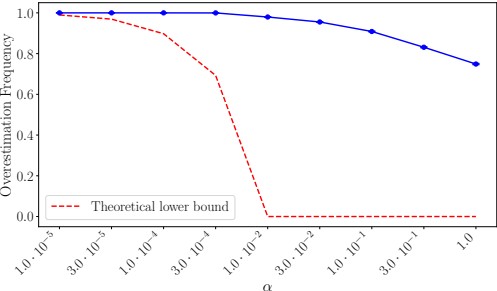 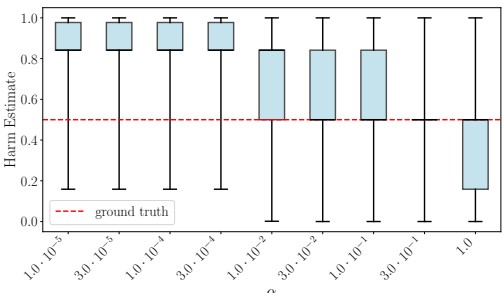

(a) The frequency with which the inequality in Prop. 4.6 is satisfied.

(b) The distribution of the right-hand side of (8), for an action with a true harm probability of 0.5.

Figure 2: Overestimate frequency and harm estimate distribution for the Prop. 4.6 guardrail for varying $\alpha$.

The bandit comes with a notion of harm: if the reward received at a given timestep exceeds some threshold $E$, the bandit explodes and the agent dies[7], terminating the episode. In other words, we define harm as $H_t := \mathbb{1}[R_t > E]$, where $R_t$ is the random variable representing the reward received when taking action $a_t$. $E$ is set to a Monte Carlo approximation of the expected highest mean reward of any action (*i.e.*, $\mathbb{E}[\max_a(f_a \cdot v^*)]$). The maximum episode length is 25 timesteps.

**Safety guardrails.** A *guardrail* is an algorithm that, given a possible action and context (*e.g.*, current state and history), determines whether taking the action in the context is admissible. A guardrail can be used to mask the policy to forbid certain actions, such as those whose estimated harm exceeds some threshold $C$. We compare several guardrails (formally defined below): those constructed from Prop. 3.4 and Prop. 4.6, one that marginalizes across the posterior over $\tau$ to get the posterior predictive harm probability, and one that 'cheats' by using the probability of harm under the true theory $\tau^*$. Recall that $Z_{1:t}$ consists of the observations (*i.e.*, actions taken and rewards received) at previous timesteps.

- **Prop. 3.4 guardrail:** rejects an action $a_{t+1}$ if there exists $\tilde{v} \in \arg\max_v \mathrm{P}(v \mid Z_{1:t}) \mathrm{P}(H_{t+1} = 1 \mid \tau, Z_{1:t}, a_{t+1})$ with $\mathrm{P}(H_{t+1} = 1 \mid \tau_{\tilde{v}}, Z_{1:t}, a_{t+1}) > C$ (note that the assumptions of i.i.d. observations and distinct theories are not satisfied here).
- **Prop. 4.6 guardrail:** rejects an action $a_{t+1}$ if $\max_{v \in \mathcal{I}_{Z_{1:t}}^\alpha} \mathrm{P}(H_{t+1} = 1 \mid Z_{1:t}, \tau_v, a_{t+1}) > C$.
- **Posterior predictive guardrail:** rejects an action $a_{t+1}$ if $\mathrm{P}(H_{t+1} = 1 \mid Z_{1:t}, a_{t+1}) > C$.
- **Cheating guardrail:** rejects an action $a_{t+1}$ if $\mathrm{P}(H_{t+1} = 1 \mid Z_{1:t}, \tau^*, a_{t+1}) > C$ (note that this guardrail assumes knowledge of the true theory $\tau^*$).

The guardrail is run at every sampling step, and actions that the guardrail rejects are forbidden to be sampled by the agent. If all actions are rejected by the guardrail, the episode terminates.

**Results.** Fig. 1 shows mean episode rewards and episode deaths under each guardrail across 10000 episodes, for different values of the rejection threshold $C$. The cheating guardrail achieves near zero deaths for sufficiently small $C$, but for $C = 0.1$ its death probability is high.[8] The posterior predictive guardrail achieves zero deaths for small $C$, while for larger $C$ it dies frequently, generally receiving lower reward compared to the cheating guardrail. The behavior of the Prop. 4.6 guardrail depends strongly on $\alpha$. When $\alpha$ is close to 1, actions are rarely rejected, leading to frequent deaths. Up to a point, this riskier behavior allows the agent to get more reward, but for $C = 0.1$ and high $\alpha$ the trend starts to reverse, as early deaths become frequent enough to preclude the opportunity. At the other extreme, when $\alpha$ is close to 0, the candidate set of theory indices $\mathcal{I}_{Z_{1:t}}^\alpha$ is larger and the guardrail is extremely conservative. It rejects almost all actions, resulting in low deaths and low reward. This is the case even for larger $C$, since the estimated probability used to filter actions tends to overestimate an action's harm probability under the true theory. For middling values of $\alpha$, Prop. 4.6 guardrail performs more similarly to the posterior predictive, sometimes with lower reward and higher deaths, and sometimes the opposite. The Prop. 3.4 guardrail, which makes the incorrect assumptions of i.i.d. data and distinct theories, is similarly conservative to the Prop. 4.6 guardrail with low $\alpha$.

**Tightness of bounds.** Fig. 2 shows how often and how tightly the inequality in Prop. 4.6 is satisfied. For an agent following a uniform policy across 10000 bandit episodes without action rejection

---

[7]This emulates in a simplified form the important and problematic scenario where the user goal, *e.g.*, maximizing profits, conflicts with safety, and we need to, for example, maximize profit under safety constraints.

[8]Indeed, if every action taken had a harm probability of 0.1, the probability of death across an episode would be $1 - ((1 - 0.1)^{25}) \approx 0.93$.

or death, Fig. 2a shows the frequency with which $\max_{v \in \mathcal{I}_{Z_{1:t}}^{\alpha}} P(H_{t+1} = 1 \mid Z_{1:t}, \tau_v, a_{t+1})$ overestimates the true harm probability. Prop. 4.6 gives us a strict lower bound of $1 - \frac{\alpha}{P(v^*)}$ (which may be below 0) on the overestimation frequency, but the frequency significantly exceeds the bound for larger $\alpha$. Fig. 2b shows the distribution of harm estimates for actions with a ground truth harm probability of 0.5. For large $\alpha$ the harm of these dangerous actions is usually *underestimated* – so the high overestimation rate in Fig. 2a comes from actions with lower harm probabilities.

## 6   CONCLUSION AND OPEN PROBLEMS

The approach to safety verification proposed here is based on context-dependent run-time verification because the set of possible inputs for a machine learning system is generally astronomical, whereas the safety of the answer to a specific question is more likely to be tractable. It focuses on the risk of wrongly interpreting the data, including the safety specification itself (called "harm" above) and exploits the fact that, as more evidence is gathered (necessary with i.i.d. data) and when different theories predict different observations, the true interpretation rises towards the maximal value of the Bayesian posterior. The bound is tighter with the i.i.d. data, but the i.i.d. assumption is also unrealistic, and for safety-critical decisions, we would prefer to err on the side of prudence and fewer assumptions. However, it provides a template to think about variants of this idea in future work. Several challenges remain for turning such bounds into an operational run-time safeguard:

1. **Upper-bounding overcautiousness.** Can we ensure that we do not underestimate the probability of harm but do not massively overestimate it? Some simple theories consistent with the dataset (even an arbitrarily large one) might deem non-harmful actions harmful. Can we bound how much this harm-avoidance hampers the agent? A plausible approach would be to make use of a mentor for the agent that demonstrates non-harmful behavior (Cohen & Hutter, 2020).

2. **Tractability of posterior estimation.** How can we efficiently estimate the required Bayesian posteriors? For computational tractability, a plausible answer would rely on amortized inference, which turns the difficult estimation of these posteriors into the task of training a neural net probabilistic estimator which will be fast at run-time. Recent work on amortized Bayesian inference for symbolic models, such as causal structures (Deleu et al., 2022; 2023), and for intractable posteriors in language models (Guo et al., 2021; Hu et al., 2024; Venkatraman et al., 2024; Song et al., 2024; Yu et al., 2024) – which are useful when prior knowledge is encoded in a pretrained foundation model – suggests that this is feasible. Advances in efficient and adaptive Monte Carlo methods, *e.g.*, for language models (Phan et al., 2023; Zhao et al., 2024; Lew et al., 2023), can also be useful for this purpose, and MCMC approaches can complement and aid amortization (Hu et al., 2023; Kim et al., 2024b; Sendera et al., 2024; Kim et al., 2024a).

3. **Efficient search for a cautious theory.** How can we efficiently identify a cautious but plausible theory that upper-bounds the risk, since this requires an optimization at run-time? Again, a plausible answer is to rely on amortized probabilistic inference, *e.g.*, by sampling theories with a low or annealed temperature, a technique that has been used for decades in the nonconvex optimization literature and more recently combined with amortized sequential sampling methods (Zhang et al., 2023a;b; Zhou et al., 2024, *inter alia*).

4. **Partial specification of theories.** Another issue is that a full explanatory theory $\tau$ for the evidence $D$ will generally be too large an object to make the above bound calculations and optimization sufficiently efficient. An alternative would be to derive bounds that only require partial explanations, similarly to how humans tell cautious 'stories' about what could go wrong in uncertain circumstances when thinking about taking a potentially dangerous decision (Bengio, 2024).

5. **Harm specification in natural language.** How can natural language safety specifications, as in Constitutional AI (Bai et al., 2022), be turned into a random variable over which conditional probabilities can be estimated? Manually labeling numerous examples matching the desired constraint is impractical. Instead, Bengio (2024) propose learning to interpret natural language statements as formal random variables over which probabilistic causal reasoning can occur. The ambiguity of natural language can then be captured by a distribution over formal interpretations.

6. **Effect of approximation error.** Finally, because we plan to use machine learning to estimate the bound, we have to consider the fact that a learned predictor will be imperfect. How do we conservatively correct for the potentially dangerous errors made by this predictor? This may be crucial if we use a neural network to estimate probabilities, since they are subject to lack of robustness and adversarial attacks (Goodfellow et al., 2015; Wei et al., 2023), *e.g.*, how do we ensure that the agent proposing the actions has not learned to find loopholes in the risk estimator?

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

## A  PROOFS

**Proposition 3.1** (**True theory dominance**). *Under the above conditions and supposing that* $P(\tau^*) > 0$, *the posterior* $P(\cdot \mid Z_{1:t})$ *is almost surely defined for all n, and the following almost surely hold:*

(a) $P(\cdot \mid Z_{1:t}) \xrightarrow{t \to \infty} \delta_{\tau^*}$ *as measures, where* $\delta_{\tau^*}$ *is the Dirac measure, which assigns mass* 1 *to the theory* $\tau^*$ *and* 0 *elsewhere; equivalently,* $\lim_{t \to \infty} P(\tau \mid Z_{1:t}) = \mathbb{1}[\tau = \tau^*]$.

(b) *There exists* $N \in \mathbb{N}$ *such that* $\arg\max_{\tau \in \mathcal{M}} P(\tau \mid Z_{1:t}) = \tau^*$ *for all* $t \geq N$.

*Proof of Proposition 3.1.* This is an application of Doob's posterior consistency theorem (Doob (1949); see also Miller (2018) for a modern summary). This result, which follows from the theory of martingales, assumes that $\tau^*$ is sampled from the prior distribution $P(\tau)$ and the observations $Z_i$ are defined as above. Doob's theorem states that if for every $S \in \mathcal{F}$, the map $\tau \mapsto P_\tau(S)$ is measurable, then the posteriors $P(\cdot \mid Z_{1:t})$ are almost surely defined and (a) holds P-almost surely with respect to the choice of $\tau^*$.

In our case, because $\mathcal{M}$ is countable, the measurability condition is satisfied, showing that (a) holds for P-almost every $\tau^* \in \mathcal{M}$. In particular, if $P(\tau^*) > 0$, then (a) holds.

Finally, by (a), we have that for any $\varepsilon > 0$, there exists $N$ such that for every $t \geq N$, $P(\tau^* \mid Z_{1:t}) > 1 - \varepsilon$, or, equivalently, $\sum_{\tau \neq \tau^*} P(\tau \mid Z_{1:t}) < \varepsilon$, and therefore $P(\tau \mid Z_{1:t}) < \varepsilon$ for all $\tau \neq \tau^*$. In particular, taking $\varepsilon = 1/2$, we get that for sufficiently large $t$, $P(\tau^* \mid Z_{1:t}) > P(\tau \mid Z_{1:t})$ for every $\tau$, which shows (b). ⊠

**Lemma 3.3.** *Under the conditions of Prop. 3.1, let* $f : \mathcal{M} \times \bigcup_{t=0}^{\infty} \mathcal{Z}^t \to \mathbb{R}_{\geq 0}$ *be a bounded measurable function. Then there exists* $N \in \mathbb{N}$ *such that for all* $t \geq N$ *and any* $\tilde{\tau} \in \arg\max_\tau [P(\tau \mid Z_{1:t}) f(\tau, Z_{1:t})]$, *it holds that* $f(\tau^*, Z_{1:t}) \leq f(\tilde{\tau}, Z_{1:t})$.

*Proof of Lemma 3.3.* First, note that the argmax exists by boundedness of $f$ and $P(\cdot \mid Z_{1:t})$. By Prop. 3.1 (b), there exists $N \in \mathbb{N}$ such that for all $t \geq N$ and $\tau \neq \tau^*$, $P(\tau^* \mid Z_{1:t}) > P(\tau \mid Z_{1:t}) \geq 0$. Let $t \geq N$ and $\tilde{\tau} \in \arg\max_\tau [P(\tau \mid Z_{1:t}) f(\tau, Z_{1:t})]$. Then

$$P(\tau^* \mid Z_{1:t}) f(\tilde{\tau}, Z_{1:t}) \geq P(\tilde{\tau} \mid Z_{1:t}) f(\tilde{\tau}, Z_{1:t}) \geq P(\tau^* \mid Z_{1:t}) f(\tau^*, Z_{1:t}).$$

When $\tilde{\tau} \neq \tau^*$, the result follows since $P(\tau^* \mid Z_{1:t}) > 0$. The case $\tilde{\tau} = \tau^*$ is trivial. ⊠

**Proposition 3.4** (**Harm probability bound**). *Under the same conditions as Prop. 3.1, there exists* $N \in \mathbb{N}$ *such that for all* $t \geq N$ *and* $\tilde{\tau} \in \arg\max_\tau P(\tau \mid Z_{1:t}) P(H = 1 \mid x, \tau, Z_{1:t})$, *it holds that*

$$P(H = 1 \mid x, \tau^*, Z_{1:t}) \leq P(H = 1 \mid x, \tilde{\tau}, Z_{1:t}). \tag{4}$$

*Proof of Proposition 3.4.* Apply Lemma 3.3 to the function $f(\tau, Z_{1:t}) = P(H = 1 \mid x, \tau, Z_{1:t})$. ⊠

**Lemma 4.1** (Martingale). *The process* $W_t := P(i^* \mid Z_{1:t})^{-1}$ *is a supermartingale,* i.e., *it does not increase over time in expectation.*

*Proof of Lemma 4.1.* We have

$$\mathbb{E}_{\tau^*}[W_{t+1} \mid Z_{1:t} = z_{1:t}] = \int_{\{z_{t+1} \in \mathcal{Z} : P_{\tau^*}(z_{t+1} \mid z_{1:t}) > 0\}} P(i^* \mid z_{1:t+1})^{-1} P_{\tau^*}(z_{t+1} \mid z_{1:t}) \, d\mu$$

$$\overset{(a)}{=} \int_{\{z_{t+1} \in \mathcal{Z} : P_{\tau^*}(z_{t+1} \mid z_{1:t}) > 0\}} \frac{\sum_{j \in I} P(j \mid z_{1:t}) P_{\tau_j}(z_{t+1} \mid z_{1:t})}{P(i^* \mid z_{1:t}) P_{\tau^*}(z_{t+1} \mid z_{1:t})} P_{\tau^*}(z_{t+1} \mid z_{1:t}) \, d\mu$$

$$\overset{(b)}{\leq} \int_{\mathcal{Z}} \frac{\sum_{j \in I} P(j \mid z_{1:t}) P_{\tau_j}(z_{t+1} \mid z_{1:t})}{P(i^* \mid z_{1:t})} \, d\mu$$

$$= w_t \sum_{j \in I} P(j \mid z_{1:t}) \int_{\mathcal{Z}} P_{\tau_j}(z_{t+1} \mid z_{1:t}) \, d\mu$$

$$\overset{(c)}{=} w_t \tag{9}$$

where $(a)$ is by the definition (5), $(b)$ follows from cancellation and positivity of the integrand, $w_t := P(i^* \mid Z_{1:t} = z_{1:t})^{-1}$ is the realization of $W_t$, and $(c)$ follows because both the posterior and the conditional probability measure integrate to 1.

This holds for any $z_{1:t}$, so we can remove the conditional:

$$\mathbb{E}_{\tau^*}[W_{t+1} \mid w_t] = \int_{z_1,\ldots,z_t} \mathbb{E}_{\tau^*}[W_{t+1}|z_{1:t}, w_t]\, P_{\tau^*}(z_1, \ldots, z_t \mid W_t = w_t)\, d\mu^{\otimes t}$$

$$\overset{(a)}{=} \int_{z_{1:t}} \mathbb{E}_{\tau^*}[W_{t+1}|z_{1:t}]\, P_{\tau^*}(z_{1:t} \mid w_t)\, d\mu^{\otimes t}$$

$$\overset{(b)}{\leq} \int_{z_{1:t}} w_t\, P_{\tau^*}(z_{1:t} \mid w_t)\, d\mu^{\otimes t}$$

$$= w_t$$

where $(a)$ follows because $w_t$ is a function of $z_{1:t}$ and $(b)$ is Inequality 9. $\boxtimes$

**Proposition 4.2** (Posterior on truth). *For all $\delta > 0$, with probability at least $1 - \delta$, $\inf_t P(i^* \mid Z_{1:t}) \geq \delta\, P(i^*)$; that is, $\tau^*\left(\{z_{1:\infty} : \inf_t P(i^* \mid z_{1:t}) < \delta\, P(i^*)\}\right) \leq \delta$, or equivalently:*

$$\tau^*\left(\sup_{t \geq 0} W_t \geq (\delta\, P(i^*))^{-1}\right) \leq \delta \tag{6}$$

*Proof of Proposition 4.2.* By Ville's inequality (Ville, 1939) for the supermartingale $W_t := P(i^* \mid Z_{1:t})^{-1}$, for any $\lambda > 0$:

$$\tau^*\left(\sup_{t \geq 0} W_t \geq \lambda\right) \leq \frac{\mathbb{E}[W_0]}{\lambda} = \frac{1}{\lambda\, P(i^*)}$$

Setting $\lambda = (\delta\, P(i^*))^{-1}$, we get

$$\tau^*\left(\sup_{t \geq 0} W_t \geq (\delta\, P(i^*))^{-1}\right) \leq \delta$$

and given that

$$\left\{z_{1:\infty} : \sup_{t \geq 0} w_t := \sup_{t \geq 0} P(i^* \mid z_{1:t})^{-1} > (\delta\, P(i^*))^{-1}\right\} = \left\{z_{1:\infty} : \inf_{t \geq 0} P(i^* \mid z_{1:t}) < \delta\, P(i^*)\right\},$$

the result follows. $\boxtimes$

**Proposition 4.4** (Weak harm probability bound). *For any $\delta > 0$, with probability at least $1 - \delta$, the following holds for all $t \in \mathbb{N}$ and all $x_t$:*

$$P(H_t = 1 \mid Z_{1:t}, \tau^*, x_t) \leq \sup_{i \in I} \frac{P(i \mid Z_{1:t})\, P(H_t = 1 \mid Z_{1:t}, \tau_i, x_t)}{\delta\, P(i^*)}.$$

*Proof of Proposition 4.4.* Substituting $i$ for $i^*$ on the r.h.s. can never increase the r.h.s., since $i^* \in I$. Then, after canceling and rearranging the terms, the proposition is readily implied by Prop. 4.2. $\boxtimes$

**Proposition 4.5** (**True theory dominance**). *If $\alpha < \delta\, P(i^*)$, then with probability at least $1 - \delta$, for all $t \in \mathbb{N}$, $i^* \in \mathcal{I}^\alpha_{Z_{1:t}}$.*

*Proof of Proposition 4.5.* For any $t \geq 1$, by Prop. 4.2,

$$\delta \geq \tau^*\left(\left\{z_{1:\infty} : \inf_{t'} P(i^* \mid z_{1:t'}) < \delta\, P(i^*)\right\}\right)$$

$$\geq \tau^*(\{z_{1:\infty} : P(i^* \mid z_{1:t}) < \delta\, P(i^*)\})$$

$$\geq \tau^*(\{z_{1:\infty} : P(i^* \mid z_{1:t}) < \alpha\}).$$

So $\tau^*(\{z_{1:\infty} : P(i^* \mid z_{1:t}) \geq \alpha\}) \geq 1 - \delta$, and the result follows by the fact that $\mathcal{I}^\alpha_{Z_{1:t}} \supseteq \{i \in I : P(i \mid Z_{1:t}) \geq \alpha\}$, since the sum in (7) never exceeds 1. $\boxtimes$

**Proposition 4.6** (**Harm probability bound**). *If $\alpha < \delta\, P(i^*)$, then with probability at least $1 - \delta$, for all $t \in \mathbb{N}$ and all $x_t$,*

$$P(H_t = 1 \mid Z_{1:t}, \tau^*, x_t) \leq \max_{i \in \mathcal{I}^\alpha_{Z_{1:t}}} P(H_t = 1 \mid Z_{1:t}, \tau_i, x_t) \tag{8}$$

*Proof of Proposition 4.6.* This follows directly from Prop. 4.5. $\boxtimes$

