# OpenReview forum: "Can a Bayesian oracle prevent harm from an agent?"
_ICLR.cc/2025/Conference — Submitted to ICLR 2025_

### Official Review · Reviewer_bi5T · 2024-10-28

**Soundness:** 4
**Presentation:** 2
**Contribution:** 2
**Rating:** 5
**Confidence:** 3

**Summary:**

The paper is tackling the problem of safety in AI. The authors take the view of defining safety as avoiding certain undesirable states in specific contexts.
They introduce a framework based on Bayesian inference from which an agent can derive safe policies that come with (probabilistic) guarantees of preventing harm.
The approach is safe-by-design, i.e. able to prevent undesired outcomes even if no concrete example of harmful states was ever observed in the system.

**Strengths:**

The main strength of the paper is the introduction of a (as far as I am aware) novel view at safe-by-design in AI at runtime and opening the possibilities for future Bayesian methods to utilize the safety guarantees shown in the paper. Allowing for safety guarantees and steering future work in a direction that empashizes those is a significant problem in AI.
I especially appreciate the discussion on open problems of the approach in the conclusion.

The theory being developed is also quite general and spans over a wide range of possible systems/problems.

The paper is well motivated and generally well structured, introducing formal concepts as needed in the respective sections. A small experimental evaluation is performed and well discussed. Proofs are provided in the appendix and I could not find any mistakes.

**Weaknesses:**

My main critique points of the paper are the lack of technical novelty (or at least it is not clarified enough if there are new results) and questions on applicability.

For the former, essentially all Propositions and Lemmata are either adaptations of well known results (Prop 3.1), taken from previous literature (Lemma 4.1, Prop 4.2), or rather simple Corollary derived from them (Lemma 3.3, Props. 3.4, 4.4, 4.5, 4.6). For Prop. 4.2 is is shown that the result is tight (Remark 4.3). To me it did not became clear whether this is a known result or a new contribution. It is also not clear whether the derived results (Props. 4.4, 4.5, 4.6) are also tight as a consequence or whether there is room for improvement.
For Prop. 4.5 and 4.6 in particular, restricting the possible world models indeces to $\mathcal{I}^{\alpha}_{1\colon t}$ is essential, however, the choice of definition of $\mathcal{I}^{\alpha}$ is not really motivated. At the same time, Fig. 2(a) shows a substantial gap between applying Prop 4.6 in practice, and the theoretical optimum. This begs the question whether a different definition of $\mathcal{I}^{\alpha}$ (e.g. a simple cutoff, or requiring $\mathcal{I}^{\alpha}$ to have a certain probability mass) has potential to yield tighter bounds. However, as the definition of $\mathcal{I}^{\alpha}$ is not motivated, these questions remain unadressed.

For the applicability, my core concern is that the main problem in AI is not providing safety guarantees under certain assumptions, but rather designing a Bayesian agent that actually works well for a given problem while satisfying these assumptions. To go into detail, section 3 only provides "law of large numbers"-style guarantees which are not useful in practice. A small paragraph on the rate of convergence (which would be very helpful to know) is included but essentially is very problem-dependent and thus not discussed in detail in this more general framework. In the experimental evaluation, where Prop. 3.4 is utilized, it is not even clear whether $t$ is large enough for the guarantee statement of Prop. 3.4 to hold (on top of Prop 3.4 not being applicable due to non-i.i.d. as the authors mention themselves). Section 4 then relaxes to probabilistic guarantees, which is a more practical approach. However, to apply the results of section 4 in practice it ultimately relies on defining a hyperparameter alpha. On the theoretical side, the guarantees in section 4 only hold if alpha is chosen small enough (which is impossible to know without knowing the system in the first place) and on the practical side, the evaluation in section 5 shows that choosing alpha too large can have catastrophic consequences, even for the simple bandit system considered in section 5. In summary, I do not see any immediate way to take advantage of the theoretical results the paper provides. This is also amplified by the fact that the main body essentially does not discuss related work, and how existing approaches can be embedded into the framework.

*These weaknesses make the paper feel like more of a statement paper with some additional mathematical background, rather than a fully fletched research paper.*

As a minor comment, from a reader's POV, the paper can be hard to follow at times, especially in the formal sections. Many paragraphs are written in a very technical way, assuming a deep mathematical background. While this surely can be expected from an audience like ICLR, I feel like many sections disrupt the flow of the paper, e.g. the two paragraphs "Setting" (l.155ff and l. 268ff). While these are defnitely important to make the paper rigorous, they are not strictly required to convey the main ideas of the paper. In the interest of readability, it might be advantageous to instead outsource the technical definitions to a separate section.

**Questions:**

1. Can you detail how you can utilize the CLT to obtain convergence rates (line 238ff)? If you applied this to the example in seciton 5, would it yield practical bounds?

2. Are the results in Propositions 4.4 to 4.6 tight (in a similar vein as Remark 4.3 shows for Proposition 4.2)?

3. How do you motivate the definition of $\mathcal{I}^{\alpha}_{1\colon t}$ and have you considered different approaches?

4. Can you provide some heuristics on choosing a safe, yet effective $\alpha$ a priori? Which information might be helpful for this from e.g. which model paramteres have the biggest impact on $\alpha$ and what information from a domain expert could be incorporated?

5. Can you make any predictions on how you proposed guardrails perform on larger, more complex models? In particular, how do you expect the overestimation of harm (see Fig. 2) to be affected?

6. Are there any existing works in which your framework fits, i.e. for which you can give (probabilistic) guarantees where they were previously unavailable?
If not, are there certain settings in which you can make reasonable a priori assumptions such that your framework is applicable and concrete guarantees can be derived for a given data set?

minor comments:
- line 96: explain what $q$ is
- line 193: introduce delta as dirac notation beforehand
- the axis and legends in the figures in section 5 are barely readable

---

> ### Author Response · Authors · 2024-11-23
>
> > Essentially all Propositions and Lemmata are either adaptations of well known results (Prop 3.1), taken from previous literature (Lemma 4.1, Prop 4.2), or rather simple Corollary derived from them (Lemma 3.3, Props. 3.4, 4.4, 4.5, 4.6). For Prop. 4.2 is is shown that the result is tight (Remark 4.3).
>
> This is substantially correct. So we think the question at stake is: can it be groundbreaking to construct a theoretical solution to a problem (the problem of harm avoidance in this case) by finding existing theoretical machinery that is fit-for-purpose and putting “wrappers” on it? We don’t mean this to be rhetorical; “no” is a valid opinion here. But a problem with “no” is that in some contexts, it means that only overcomplicated solutions to a problem will make it to ICLR, or no solutions at all. In defense of “yes”, simplicity is a virtue. If we provide a valid solution to an important problem, but it is not complicated enough, that hardly seems bad.
>
> The argument we just provided would fail if the average ML theorist or statistician would quickly reproduce our results when posed the question, “How can an agent lower bound harm probability in theory?”. Then it wouldn’t be a problem if no solutions make it to ICLR. But we don’t think this is the case.
>
> Regarding Remark 4.3, we are not aware of this result having been proven before.
>
> > It is also not clear whether the derived results (Props. 4.4, 4.5, 4.6) are also tight as a consequence or whether there is room for improvement. For Prop. 4.5 and 4.6 in particular, restricting the possible world models indeces to I1:tα is essential, however, the choice of definition of Iα is not really motivated. At the same time, Fig. 2(a) shows a substantial gap between applying Prop 4.6 in practice, and the theoretical optimum. This begs the question whether a different definition of Iα (e.g. a simple cutoff, or requiring Iα to have a certain probability mass) has potential to yield tighter bounds. However, as the definition of Iα is not motivated, these questions remain unadressed.
> >
> > Q3. How do you motivate the definition of I1:tα and have you considered different approaches?
>
> These results are not tight, unfortunately, but we conjecture that we cannot achieve tight bounds in general without making unreasonable assumptions.
> Despite that, we did conduct experiments to validate this $I^α_{Z_{1:t}}$ and the corresponding Prop 4.6 bound empirically in the context of our bandits experiment, comparing them against the minimal non-empty set of indices that would satisfy Prop 4.5 (defined as the union of one theory maximizing the posterior and all theories with posterior probability $≥ α$), as well as various other bounds. With this modified index set of theories, we tested various approaches to aggregating harm estimates, including different quantiles, weighted means (arithmetic/geometric/harmonic) with different power exponents, and various weighting schemes based on posterior probabilities.
> These experiments showed that our formulation achieves a good empirical trade-off between safety and performance. Another advantage of the Prop 4.6 bound in the paper is that it provides a theoretical guarantee to overapproximate the probability of harm, contrary to the alternative aggregation methods we tested, which, while sometimes achieving higher rewards, lacked such guarantees (so it came at the cost of more deaths for the agent). Due to space constraints and to maintain focus on the main contribution, we didn't include these additional experimental results in the paper (but we would be happy to add them to the appendix if you think that they would be valuable to empirically motivate $I^α_{Z_{1:t}}$ and the Prop 4.6 bound).
> On the theoretical side, the motivation for our specific definition is that if we form a Bayes mixture of just the top $n$ models, we can bound the lifetime prediction errors that this mixture makes on observed data $Z_{1:\infty}$. Likewise for the Bayes mixture of just top $n-1$ models. We can translate this into a bound on the prediction errors that model $n$ makes on its own, to the extent that model $n$ has posterior weight within the Bayes mixture of the top $n$ models. So we enforce that this weight is above $\alpha$. This reasoning was originally developed by [Cohen and Hutter (2022)](https://jmlr.org/papers/volume23/21-0618/21-0618.pdf), and it results in their Theorem 6 (i), which bounds the lifetime prediction error that any model in $I_{Z_{1:t}}^\alpha$ makes on the data $Z_{1:\infty}$. Note, however, that since harm probabilities do not necessarily appear themselves in the “training data” $Z_{1:\infty}$, it does not follow that lifetime prediction error on harm probability is bounded.

---

> ### Author Response · Authors · 2024-11-23
>
> > For the applicability, my core concern is that the main problem in AI is not providing safety guarantees under certain assumptions, but rather designing a Bayesian agent that actually works well for a given problem while satisfying these assumptions. To go into detail, section 3 only provides "law of large numbers"-style guarantees which are not useful in practice. A small paragraph on the rate of convergence (which would be very helpful to know) is included but essentially is very problem-dependent and thus not discussed in detail in this more general framework. In the experimental evaluation, where Prop. 3.4 is utilized, it is not even clear whether t is large enough for the guarantee statement of Prop. 3.4 to hold (on top of Prop 3.4 not being applicable due to non-i.i.d. as the authors mention themselves). Section 4 then relaxes to probabilistic guarantees, which is a more practical approach. However, to apply the results of section 4 in practice it ultimately relies on defining a hyperparameter alpha. On the theoretical side, the guarantees in section 4 only hold if alpha is chosen small enough (which is impossible to know without knowing the system in the first place) and on the practical side, the evaluation in section 5 shows that choosing alpha too large can have catastrophic consequences, even for the simple bandit system considered in section 5. In summary, I do not see any immediate way to take advantage of the theoretical results the paper provides.
>
> The paper explicitly states that the theoretical results only open the door to a possible direction for AI safety and that at least five challenges remain in order to turn this kind of approach into efficient and reliable guardrails. The paper is not claiming otherwise and in our opinion should be considered for its value in highlighting the theoretical basis for future work in AI safety and conservative guardrails.
>
> It is true that the bounds of Section 3 are asymptotic in data, whereas ultimately we aspire to have non-asymptotic safety bounds. Section 4 explores one way to achieve that, and we are currently investigating others. This paper is a first step.
>
> > This is also amplified by the fact that the main body essentially does not discuss related work, and how existing approaches can be embedded into the framework.
>
> We will add a number of related works to the paper, as suggested by the reviewers.
>
> > As a minor comment, from a reader's POV, the paper can be hard to follow at times, especially in the formal sections. Many paragraphs are written in a very technical way, assuming a deep mathematical background. While this surely can be expected from an audience like ICLR, I feel like many sections disrupt the flow of the paper, e.g. the two paragraphs "Setting" (l.155ff and l. 268ff). While these are defnitely important to make the paper rigorous, they are not strictly required to convey the main ideas of the paper. In the interest of readability, it might be advantageous to instead outsource the technical definitions to a separate section.
>
> Thanks for the suggestion regarding technical material. We agree that making as much of the paper as possible accessible to a broader audience of ICLR researchers is important.
>
> Regarding larger models, it is difficult to anticipate how the proposed approaches would perform or larger models because in our opinion there are several technical and algorithmic questions that need to be addressed (the five listed in the last section), which would result in methods that have the same spirit as what we proposed but probably a quite different algorithmic form.
>
> > Q1. Can you detail how you can utilize the CLT to obtain convergence rates (line 238ff)? If you applied this to the example in seciton 5, would it yield practical bounds?
>
> Convergence rates are held up by the model closest to $\tau^*$ (smallest $D_{KL}(\tau^* || \tau)$). We are interested in the sample mean of the quantity $D_{\tau}^{t+1} - D^t_{\tau}$. The expectation is positive, since it is a KL divergence, so the sample mean will converge to a positive number at a rate governed by the CLT.
>
> > Q2. Are the results in Propositions 4.4 to 4.6 tight (in a similar vein as Remark 4.3 shows for Proposition 4.2)?
>
> Unfortunately not. In general, the harm probability predictions are like off policy predictions in RL--we are not assuming that it is possible for the data $Z_{1:\infty}$ to contain direct observations of the harm probability. In this regime, tight bounds do not appear possible without making assumptions that are unlikely to be founded in practice.

---

> ### Author Response · Authors · 2024-11-23
>
> > Q4: Can you provide some heuristics on choosing a safe, yet effective α a priori? Which information might be helpful for this from e.g. which model paramteres have the biggest impact on α and what information from a domain expert could be incorporated?
>
> Intuitively, one might consider the number of qualitatively different models of harm that one could make a case for, which wouldn’t be easily ruled out by the data. Calling this number $n$, if after we get lots of data, those $n$ models will plausibly be in the top set along with the truth, and $\alpha^{-1}$ should be $O(n)$. A domain expert with an understanding of the setting and the data generating process could help make such an assessment.
>
> > Q5. Can you make any predictions on how you proposed guardrails perform on larger, more complex models? In particular, how do you expect the overestimation of harm (see Fig. 2) to be affected?
>
> It is difficult to anticipate how the proposed approaches would perform or larger models because in our opinion there are several technical and algorithmic questions that need to be addressed (the five listed in the last section), which will result in methods that have the same spirit as what we proposed but probably a quite different algorithmic form.
>
>
> > Q6. Are there any existing works in which your framework fits, i.e. for which you can give (probabilistic) guarantees where they were previously unavailable? If not, are there certain settings in which you can make reasonable a priori assumptions such that your framework is applicable and concrete guarantees can be derived for a given data set?
>
> For this framework to work robustly in realistic settings, we unfortunately await further work on computing reliable approximations of the posterior. But suppose we pretended that haphazard posterior approximations were robust, such as ensembles (see discussion in Wilson and Izmailov (2022)). Then we could apply our framework, although we wouldn’t confidently call these a priori assumptions “reasonable”. In that setting, our work lends credence to the practice of considering the worst-case within an ensemble.
>
> However, because of the challenges we listed in the end, and in particular the challenge of approximating the posterior over large theories, we do not feel that it is feasible to go beyond small-size settings like those studied in the experimental section.
>
> > minor comments: line 96: explain what q is; line 193: introduce delta as dirac notation beforehand; the axis and legends in the figures in section 5 are barely readable
>
> You’re right. Thanks for asking about $q$, we will clarify that it is an estimator of the true posterior. We will also introduce $\delta$ as a notation for Dirac before using it, and we will redo the figures in section 5 to make them more readable.

---

### Official Review · Reviewer_ADYh · 2024-11-03

**Soundness:** 3
**Presentation:** 3
**Contribution:** 2
**Rating:** 6
**Confidence:** 2

**Summary:**

This paper studies the problem of evaluating an unknown world model from observed data to determine whether it satisfies a certain safety metric. The safety metric, or guardrail, is a binary variable $H$, taking other variables in the world model as input. The authors utilize a Bayesian approach. It assumes access to the actual prior distribution over the groundtruth world model. The authors first prove that under certain parametric assumptions, the posterior distribution over candidate models will uniquely converge to the ground-truth model at the limit of large samples. Building on this concentration results, the authors derive an upper bound over the posterior probability of the harmful event $H = 1$ conditioning on the observed data. This concentration bound is then extended to non-i.i.d settings where observed samples are correlated. Finally, simulations were performed, and results supported the proposed theory.

**Strengths:**

- The paper is well-organized and clearly written. All the theoretical assumptions have been stated.
- The proposed concentration results seem reasonable. The derivations seem technically sound.
- Training large AI systems to satisfy certain safety criteria (i.e., with guardrails) is an exciting problem. This paper formulates this problem as a hypothesis-testing problem and presents non-trivial algorithms to perform the test. This problem formulation could be inspiring for other AI researchers across domains.

**Weaknesses:**

- The concentration result in Prop. 3.1 assumes "all theories in $M$ are distinct as probability measures." This assumption does not seem to hold many common probabilistic models. For instance, in the linear component analysis, the number of independent components is generally not uniquely discernible (i.e., not identifiable) with non-linear mixing functions. Also, the number of latent components in Gaussian mixtures is generally not identifiable from the observed data. This seems to suggest that the application of the proposed concentration results might be limited.
- The proposed concentration results also assume access to the actual prior distribution generating the ground-truth world model. It is unclear whether the upper bound could still hold when the prior distribution is unknown and misspecified.
- Other concentration bounds exist over the target estimates using Baysian methods. Generally, one should be able to translate empirical concentration bounds to the Bayesian settings. For instance, (Osband & Van Roy, ICML'17) translates the concentration bounds for online reinforcement learning to Bayesian regret. How does the proposed method compare to other related work? This paper should include a section discussing related work in large deviation theory and how this paper is situated in the existing literature.

- Reference: _"Osband, Ian, and Benjamin Van Roy. "Why is posterior sampling better than optimism for reinforcement learning?." International conference on machine learning. PMLR, 2017."_

**Questions:**

1. How does the upper bound in Prop. 3.4 apply if the prior distribution $P$ is misspecified?
2. How does this work compare to the existing literature on concentration bounds in the Bayesian setting? For instance, these methods could include analysis of Bayesian regret in RL, and PAC Bayes.

---

> ### Author Response · Authors · 2024-11-23
>
> > Prop. 3.1 assumes "all theories in M are distinct as probability measures." This assumption does not seem to hold many common probabilistic models.
>
> Our formalism allows for countably many discrete models. While it would be unwieldy in practice, the model classes you discuss could be replaced with a model class that is dense in them, and any duplicates could simply be removed. Since we are just giving formal results, not proposing a construction, the impracticality is not particularly problematic.
>
> Also, it is important to note that the “theories must be distinct” assumption is a limitation of the i.i.d. setting only (Section 3). However, our non-i.i.d. results (Section 4) do not require theories to be distinct as probability measures. This more general setting can handle cases like the ones you may be thinking of, where multiple parameterizations might lead to the same distribution. In fact, our experimental results use this more general setting, since in the bandit environment, multiple reward weight vectors can represent the same collection of reward distributions.
>
> > This seems to suggest that the application of the proposed concentration results might be limited.
>
> While our theoretical results make specific assumptions, we've tested their practical utility in our bandit setting through experiments comparing our approach against alternatives. We've experimented with a minimal set of indices satisfying Prop 4.5 (one theory maximizing the posterior plus those with posterior $≥ α$) along with various aggregation methods  to estimate the harm (different quantiles, weighted means (arithmetic/geometric/harmonic) with different power exponents, various weightings based on posterior probabilities). These experiments showed that our formulation achieves good safety-performance trade-offs in practice (while some alternatives occasionally achieved higher rewards, they lacked theoretical guarantees and led to more deaths), and suggests practical applicability. Due to space constraints, these additional comparison results weren't included, but could be added to an appendix.
>
> > The proposed concentration results also assume access to the actual prior distribution generating the ground-truth world model. It is unclear whether the upper bound could still hold when the prior distribution is unknown and misspecified.
>
> In our view, there is no notion of “truth” to the prior: it represents the beliefs before seeing the data. The only thing that matters asymptotically is if the correct world model has a positive probability under the prior (and of course the larger the better) and that can be obtained if the priors are non-parametric (as we think they should). In practice and with large-scale large-data applications in mind, we anticipate that a good choice of prior is fairly agnostic and flat, with exponential decay of probabilities as a function of some description length of the theories but otherwise a uniform assignment of probability mass (in a discrete space). If the prior is parametric and does not cover the correct theory then of course Bayesian posteriors can be extremely wrong, which is not acceptable in the context of safety guardrails, and we will point that out in the paper.
>
> > Q1. How does the upper bound in Prop. 3.4 apply if the prior distribution P is misspecified?
>
> If by misspecified you mean that the correct theory is not covered by the prior, then the theorems do not apply, by definition. If, instead, the concern is about whether the prior needs to match some "ground truth" distribution, the key insight is that the prior simply represents initial beliefs, and our bounds hold as long as these beliefs assign positive probability to the correct theory (cf. our previous response, just above).
>
> > Other concentration bounds exist ... for instance, (Osband & Van Roy, ICML'17) translates the concentration bounds for online reinforcement learning to Bayesian regret.
> >
> > Q2. How does this work compare to the existing literature on concentration bounds in the Bayesian setting? For instance, these methods could include analysis of Bayesian regret in RL, and PAC Bayes.
>
> We would characterize Osband and Van Roy as studying how to translate concentration bounds in a pure predictive setting into an MDP setting where exploration is necessary. We don’t propose an exploration strategy to get more information about harm probability. In the iid setting, there is no such thing as exploration, and in the non-iid setting, it’s not clear how to ensure in a general setting that the exploration isn’t itself harmful. So the work of Osband and Van Roy and others on the question of translating concentration bounds into the RL setting seems to be addressing a question that is orthogonal to ours. We will make this comparison in the paper to better situate our work.

---

### Official Review · Reviewer_pbrg · 2024-11-03

**Soundness:** 3
**Presentation:** 2
**Contribution:** 3
**Rating:** 5
**Confidence:** 3

**Summary:**

This paper studies the problem of bounding the probability of some event in the context of an unknown but consistent distribution and a Bayesian setting.

The paper is motivated by the prevention of harm by AI agents. In short, harm is inherently unavoidable since in real applications we have no direct access to the distribution governing the environment. However, if we assume a fixed distribution, and a prior assumption of that distribution, we can get better and better approximations when data is presented to us, by using the data to update our prior knowledge of the distribution. With this, we can theoretically bound the probability of doing harm. In deployment, actions whose probability of harm is larger than some threshold can be blocked.

The paper explores two cases: incoming data as iid and non iid, and obtains bounds on the probability of harm in both cases.

The paper presents an experimental evaluation on a multi-armed bandits example, blocking actions that are considered unsafe according to the different bounds obtained as well as a baseline (with an unrealistic assumption of the underlying model). The paper ends with a discussion of the open problems still to be solved to be able to use this method as a reliable guardrails for AI agents.

**Strengths:**

S1. The topic of AI safety is timely and relevant for ICLR.

S2. The theoretical results (as far as I could check) are sound.

S3. The experimental evaluation serves to showcase how these bounds could be used in a realistic scenario.

**Weaknesses:**

W1. I understand the appeal to frame this work in the context of harm by an AI agent, and I think it is an interesting point. However, there is nothing inherent to "harm" in the concept presented. The concept of "harm" could be substituted by "reward at a state" and we could be discussing the same results in a different light. I think the paper may benefit from a more general motivation.

W2. While the experimental evaluation is welcome, it is a very simple example, and one wonders if these theoretical bounds would find applicability in problems that are more complex and close to the real applications of guardrails.

W3. The concept of guardrails presented here, as an algorithm that blocks an action if it shows an expected harm larger than some threshold, is very similar to the concept of probabilistic shielding in MDPs [1] (which is essentially the "cheating guardrail in Sec. 5), and this can be extended to partially observable MDPs to eliminate the (unrealistic) assumption of having full knowledge of the ground truth [2]. The paper would benefit from comparing to these methods, especially with [2].

W4. The paper does not engage in some recent work on defining harm in similar scenarios, see for example [3] or [4]. It could be useful to understand, in light of different definitions of harm, whether the results are specific to harm prevention, or can be framed in a more general understanding of bounds over rewards.



OTHER (MINOR) REMARKS

R1. The paper is mathematically dense and difficult to follow in parts. I'm not sure whether this is a weakness on its own, but I have the feeling that the ideas conveyed are simpler than the dense mathematical presentation seems to suggest.


REFERENCES

[1] N. Jansen et al. Safe Reinforcement Learning Using Probabilistic Shields. CONCUR 2020.

[2] S. Carr et al. Safe Reinforcement Learning via Shielding under Partial Observability. AAAI 2024.

[3] S. Beckers et al. Quantifying Harm. IJCAI 2023.

[4] J. G. Richens. Counterfactual harm. NeurIPS 2022.

**Questions:**

Q1. How do you envision these guardrails to be applied in realistic scenarios? For example, consider the situation of a language model trying to obtain your passwords, or an autonomous car trying to crash with another vehicle. Could this notion of harm be applied efficiently to these realistic scenarios?

Q2. How sensitive are the results to the choice of priors in the Bayesian framework? Can the authors discuss the robustness of the proposed approach under different prior choices?

---

> ### Author Response · Authors · 2024-11-23
>
> > W1. I understand the appeal to frame this work in the context of harm by an AI agent, and I think it is an interesting point. However, there is nothing inherent to "harm" in the concept presented. The concept of "harm" could be substituted by "reward at a state" and we could be discussing the same results in a different light. I think the paper may benefit from a more general motivation.
>
> That is true, the bounds could in principle be used for other purposes, but bounds on harm probability are the motivation for both the theory and the experiments. Bounds being bounds, they will overestimate the true probabilities, which may not be generally useful, but it is still very useful in the context of safety, when tail risks can be unacceptable and we are willing to construct conservative decision rules. On top of that, there is a definitional aspect: since these bounds are used as guardrails to prevent actions that trigger $H=1$ events in the paper, $H=1$ inherently represents outcomes we want to avoid, making "harm" an appropriate framing.
> On another note, it might be good to note that we do consider the "reward at a state" case you mention in our bandit example, where, in this setting, a reward that would be too high is considered harmful.
> Granted, the generality of the mathematical framework allows it to be applied to other settings, and thanks for pointing this out: we will clarify that in the paper. However, the motivation and intended application (in the scope of this paper) is about preventing harmful outcomes in AI systems.
>
>
> > While the experimental evaluation is welcome, it is a very simple example, and one wonders if these theoretical bounds would find applicability in problems that are more complex and close to the real applications of guardrails.
>
> Much more work is needed (as pointed out at the end) to turn the proposed bounds into methods that can scale to complex problems. It is both about the nature of the bounds themselves (which involve full world models as objects to maximize over) and about the challenge of designing tractable approximations of the Bayesian posterior. However, given the high stakes in AI safety as we move towards AGI, it is really important to make such theoretical steps in order to guide the research agenda towards conservative decision-making in which safety guardrails are put in place around powerful AI systems.
>
>
> > The concept of guardrails presented here, as an algorithm that blocks an action if it shows an expected harm larger than some threshold, is very similar to the concept of probabilistic shielding in MDPs \[1\] (which is essentially the "cheating guardrail in Sec. 5), and this can be extended to partially observable MDPs to eliminate the (unrealistic) assumption of having full knowledge of the ground truth \[2\]. The paper would benefit from comparing to these methods, especially with \[2\].
>
> Thanks for pointing this out. We will add those references and how they relate to our work.
>
> > The paper does not engage in some recent work on defining harm in similar scenarios, see for example \[3\] or \[4\]. It could be useful to understand, in light of different definitions of harm, whether the results are specific to harm prevention, or can be framed in a more general understanding of bounds over rewards.
>
> Thanks for sharing these papers, which address different aspects of harm quantification but should certainly be mentioned in our paper, and we will do that. Note that the non-iid bounds in our paper are relevant to the scenario of distributional shift.
>
>
> > How do you envision these guardrails to be applied in realistic scenarios? For example, consider the situation of a language model trying to obtain your passwords, or an autonomous car trying to crash with another vehicle. Could this notion of harm be applied efficiently to these realistic scenarios?
>
> In order to apply the ideas in our paper in realistic scenarios where the world model is not trivially small, we provided five research directions in the last section of our paper. An important step is to go from manipulating full world models in the bound calculation to being able to focus on only a subset of the random variables, and we are currently working on this question, where it is enough to sample “dangerous scenarios” of how harm could occur in order to get a bound. Other challenges include the fact that “harm” is usually not directly and explicitly observed in natural data (like text or videos) but instead is a latent variable that explains some of the words or images in the data, and that we wish to be able to informally define harm in natural language. Another challenge is of course the computationally efficient estimation of the Bayesian probabilities themselves, and we have ideas on how to use recent advances in amortized probabilistic inference to get such efficient approximations.

---

> ### Author Response · Authors · 2024-11-23
>
> > How sensitive are the results to the choice of priors in the Bayesian framework? Can the authors discuss the robustness of the proposed approach under different prior choices?
>
> Like with Bayesian approaches in general, you are right to point out the importance of priors, especially in a small data regime. However, our long-term goal is to develop safety guardrails for AGI-level models, i.e., trained with very large quantities of data. In that regime, it makes sense to use very agnostic and thus very flat priors which are uniform for a given description length and then decay exponentially when multiple description lengths are possible for a variable (such as strings representing formulae or programs). The place where prior knowledge then comes in is in the language used to compute description length, and a sensible approach to this may be to use existing human-crafted languages (like the languages of mathematics and programming) for this purpose.

---

### Official Review · Reviewer_n2RR · 2024-11-05

**Soundness:** 3
**Presentation:** 3
**Contribution:** 2
**Rating:** 5
**Confidence:** 4

**Summary:**

This paper explores the problem of designing AI systems that satisfy probabilistic safety guarantees. Within a Bayesian framework and given the safety specifications (as a probability), the authors provide risk bounds for potentially harmful decisions, showing that the probability of harm can be upper-bounded by a probability that can be estimated by approximating Bayseian posterior over theories given the observed data. They study two settings: i.i.d case and non i.i.d case and provide a simple experiment to evaluate the performance of safety guardrails.

**Strengths:**

This is a very well written paper, and it is easy to follow.

The proposed approach represents a promising initial step toward designing AI systems that ensure safety through built-in probabilistic guarantees, rather than relying solely on external safety mechanisms.

The authors also outline several open problems for future work.

**Weaknesses:**

The authors present an upper bound on the harm probability, though it appears to be highly conservative. It would be valuable if they could offer a convergence rate or practical guarantees to make the framework more usable. Additionally, it is unclear how this approach compares to other conservative methods for preventing harm.

Since the theoretical results lack practical assurances, I would have appreciated more experimental validation, especially in complex and realistic settings.

Obtaining a Bayesian oracle could be very challenging (posterior distribution).

Overall, while the paper introduces a promising method for designing safer AI systems, it would greatly benefit from additional components (both theoretical and experimental) before publication.

**Questions:**

None.

---

> ### Author Response · Authors · 2024-11-23
>
> > This is a very well written paper, and it is easy to follow.
>
> Thank you for saying so.
>
> > The authors present an upper bound on the harm probability, though it appears to be highly conservative. It would be valuable if they could offer a convergence rate or practical guarantees to make the framework more usable.
>
> In the i.i.d. case, results about the rate require further assumptions about the model class, so we simply provide an outline of how to achieve them. In our general setting, Equation 3 is the tightest claim about rates we can make. Note that convergence rates are not relevant for the non-iid section because in that case we cannot expect to (safely) achieve convergence even in the limit. See for example [this paper](https://ieeexplore.ieee.org/document/9431093) on why in the RL case, wrongly assuming ergodicity (and thus convergence) can be catastrophic.
>
> > Since the theoretical results lack practical assurances, I would have appreciated more experimental validation, especially in complex and realistic settings.
> > Obtaining a Bayesian oracle could be very challenging (posterior distribution).
>
> To validate our theoretical results empirically in the context of our bandit experiment, we did conduct additional experiments comparing our bound against alternatives. We tested a minimal set of indices that would satisfy our theoretical requirements (one theory maximizing the posterior plus those with posterior $≥ α$) along with various aggregation methods (different quantiles, weighted means, etc.). This revealed that our formulation achieves good empirical trade-offs between safety and performance. Another important aspect is that our bound provides theoretical guarantees to overapproximate the harm probability, while alternatives that sometimes achieved higher rewards did so at the cost of more deaths. Due to space constraints, we didn't include these additional comparison results in the paper, but would be happy to add them to the appendix.
> On another note, it should be noted that taking this approach a step further and turning the proposed bounds into scalable algorithms is not trivial and will require many more years of research, which is why the paper indicates at the end several needed research directions in order to achieve this. Hence, only small-scale experiments can be done at this point, and have been done and described in the paper, to at least validate the ideas and mathematical results in the paper, showing how the bounds can be used to deliver safer decision-making in a bandit scenario. Indeed, we hope the results in this paper will help recruit more attention to the problem of modelling a Bayesian posterior in complex and realistic settings.

---

### Meta-Review · Area_Chair_M29S · 2024-12-21

**Metareview:**

This paper seeks to create practical safety mechanisms for AI by introducing a Bayesian framework.
The reviewers agreed that the topic is important, given the potential implications for AI safety (e.g., Reviewer bi5T noted that the framework tackles “a significant problem in AI”) However, the reviewers were in agreement on certain limitations:
- Reviewer n2RR commented on the bounds being “highly conservative”, and questioned their practical applicability. Similarly, Reviewer bi5T  wrote that they “[..] do not see any immediate way to take advantage of the theoretical results the paper provides.”
- On a similar note again, Reviewer pbrg seemed skeptical about the paper's ability to generalize to problems “[..] complex and close to [..] real applications.”

Based on the agreement around these critiques, the decision was made to reject the paper. While the theoretical contributions were considered sound, expanding on what the theoretical model can offer in practice might be warranted.

**Additional Comments On Reviewer Discussion:**

The main concerns raised by the reviewers centered around the applicability of the framework in practice, i.e., beyond the theoretical guarantees (see above for a short summary). The authors counterargued that the value of the paper lies in the theoretical groundwork. The reviewers did not change their position that more validation is in order.

In addition to the above, some reviewers suggested citing additional relevant literature, including probabilistic shielding in MDPs and Bayesian regret in reinforcement learning.

---

### Decision · Program_Chairs · 2025-01-22

Reject